# Magnetic Nanocomposite Materials Based on Fe_3_O_4_ Nanoparticles with Iron and Silica Glycerolates Shell: Synthesis and Characterization

**DOI:** 10.3390/ijms241512178

**Published:** 2023-07-29

**Authors:** Tat’yana G. Khonina, Alexander M. Demin, Denis S. Tishin, Alexander Yu. Germov, Mikhail A. Uimin, Alexander V. Mekhaev, Artem S. Minin, Maxim S. Karabanalov, Alexey A. Mysik, Ekaterina A. Bogdanova, Victor P. Krasnov

**Affiliations:** 1Postovsky Institute of Organic Synthesis, Russian Academy of Sciences (Ural Branch), 620990 Ekaterinburg, Russia; khonina@ios.uran.ru (T.G.K.); mehaev@ios.uran.ru (A.V.M.); ca@ios.uran.ru (V.P.K.); 2Mikheev Institute of Metal Physics, Russian Academy of Sciences (Ural Branch), 620990 Ekaterinburg, Russia; germov@imp.uran.ru (A.Y.G.); uimin@imp.uran.ru (M.A.U.); calamatica@gmail.com (A.S.M.); a.mysik@ya.ru (A.A.M.); 3Institute of New Materials and Technologies, Ural Federal University, 620002 Ekaterinburg, Russia; m.s.karabanalov@urfu.ru; 4Institute of Solid State Chemistry, Russian Academy of Sciences (Ural Branch), 620990 Ekaterinburg, Russia; chemi4@rambler.ru

**Keywords:** magnetic nanoparticles, Fe_3_O_4_, glycerolates, Mössbauer spectroscopy, alternating magnetic field

## Abstract

Novel magnetic nanocomposite materials based on Fe_3_O_4_ nanoparticles coated with iron and silica glycerolates (MNP@Fe(III)Glyc and MNP@Fe(III)/SiGlyc) were obtained. The synthesized nanocomposites were characterized using TEM, XRD, TGA, VMS, Mössbauer and IR spectroscopy. The amount of iron and silica glycerolates in the nanocomposites was calculated from the Mössbauer spectroscopy, ICP AES and C,H-elemental analysis. Thus, it has been shown that the distribution of Fe in the shell and core for MNP@Fe(III)Glyc and MNP@Fe(III)/SiGlyc is 27:73 and 32:68, respectively. The synthesized nanocomposites had high specific magnetization values and a high magnetic response to the alternating magnetic field. The hydrolysis of shells based on Fe(III)Glyc and Fe(III)/SiGlyc in aqueous media has been studied. It has been demonstrated that, while the iron glycerolates shell of MNP@Fe(III)Glyc is resistant to hydrolysis, the silica glycerolates shell of MNP@Fe(III)/SiGlyc is rather labile and hydrolyzed by 76.4% in 24 h at 25 °C. The synthesized materials did not show cytotoxicity in in vitro experiments (MTT-assay). The data obtained can be used in the design of materials for controlled-release drug delivery.

## 1. Introduction

Magnetic nanoparticles (MNPs) based on iron oxides (Fe_3_O_4_, γ-Fe_2_O_3_, etc.) due to their unique properties (primarily magnetic), the possibility of varying sizes and shapes, the ease of their surface modification as well as biocompatibility are widely used in various fields of science and technology: catalysis [1], biomedicine [2,3,4,5], food safety monitoring [6] environmental remediation and energy [7,8,9], etc. Currently, there is a wide range of methods to obtain MNPs, which make it possible to synthesize particles of various sizes and shapes [10]. The group of chemical methods includes coprecipitation [11], thermal decomposition [12], solvothermal [13], hydrothermal [14], polyol [15], sol–gel [16], extraction–pyrolytic [17] and some other methods. It should be noted that the magnetic properties of particles strongly depend on the particle size [18]. Therefore, each direction of use of MNPs will have its own optimal particle size. For example, the presence of a 5–20 nm magnetic core in core–shell MNPs makes it possible to observe their distribution in the body using magnetic resonance imaging (MRI) [19,20,21,22,23] or magnetic particle imaging (MPI) [24,25], localize them in the required place using magnetic probes [26] as well as also heat them up when exposed to an alternating magnetic field, causing a hyperthermic effect [27,28,29,30,31]. Often, MNPs are also used for biological purposes, for example, for cell labeling, cell tracking and targeting for tissue engineering approaches [32,33,34,35,36] in cell [37] or molecular [38,39] separation. As the core, nanoparticles based on iron oxides (magnetite, maghemite, ferrites) are most often used, which have pronounced magnetic properties, low toxicity and biocompatibility [2].

The development of new methods for applying various types of coatings to MNPs for biomedical applications has been a widespread topic of scientific research in recent decades and is widely represented in the scientific literature. The shell in such systems primarily plays a stabilizing role, increases their stability in the physiological environment, and improves biocompatibility, biokinetics and biodistribution in the body. In addition, it allows for the adsorption of various drugs for their further use in drug or gene delivery systems [40,41,42]. The application of stimuli-responsive coatings capable of releasing the drug under certain tumor microenvironment conditions (for example, at a low pH) [43,44,45,46,47,48] or under some external influence (magnetic field, laser) is rather often used for these purposes [30,31,49,50]. MNPs with slow dissolving shells can be used as sustained-release drug delivery vehicles.

Glycerol is a commercially available, cheap, biocompatible and biodegradable polyhydric alcohol that is widely used in biomedicine. Glycerol is relatively poorly adsorbed on MNP surface. However, as we have previously demonstrated, Fe_3_O_4_ nanoparticles can be coated with iron glycerolate [51,52].

Metal glycerolates are currently used as materials in different energy technologies [53,54,55,56] as precursors for obtaining nanoparticles of various compositions by the thermal decomposition method, including MNPs, as well as various materials for technical and biomedical purposes [57,58,59,60,61].

Previously, we synthesized and characterized individual iron(III) monoglycerolate (FeC_3_H_5_O_3_, Fe(III)Glyc) for the first time [51]. Along with silicon tetraglycerolate Si(C_3_H_7_O_3_)_4_, Fe(III)Glyc was used as a biocompatible precursor in the synthesis of bioactive silicon–iron glycerolate hydrogel, which exhibits a pronounced hemostatic (feature of Fe(III)Glyc) and reparative (feature of silicon glycerolates) action [62].

It was of interest to modify the surface of MNPs based on Fe_3_O_4_, along with Fe(III)Glyc, also silica glycerolates (SiGlyc), i.e., to create a mixed Fe/Si–glycerolate (Fe(III)/SiGlyc) shell on MNPs. We assume that the mixed shell of iron and silicon glycerolates in such modified particles would not only play the role of a sorbent for drugs, but also have an additional positive pharmacological hemostatic and reparative effect, which is especially important in the oncotherapy of intracavitary organs.

Thus, the aim of this work is to develop a method for the synthesis of nanocomposites based on MNPs Fe_3_O_4_ with a shell based on Fe(III)Glyc or Fe(III)/SiGlyc, evaluate the chemical composition of the shells, reveal the features of their hydrolysis in an aqueous and aqueous–glycerol medium as well as to study the cytotoxicity of synthesized nanocomposite materials in in vitro experiments.

## 2. Results and Discussion

### 2.1. Synthesis and Characterization of Nanocomposite Materials

In this work, core–shell MNPs were synthesized with a core based on an Fe_3_O_4_ and Fe(III)Glyc shell (MNPs **1**, Figure 1a) or an Fe(III)/SiGlyc shell (MNPs **2**, Figure 1b) (Figure 1). The initial MNPs were obtained by coprecipitation by analogy with [22,63]. MNPs **1** were synthesized by heating the initial MNPs in glycerol at 180 °C for 18 h, by analogy with [51]. MNPs **2** were obtained in a similar way, but by heating of MNPs in glycerol with preliminarily synthesized silicon glycerolates of the formal composition Si(C_3_H_7_O_3_)_4_·6C_3_H_8_O_3_.

Particles were isolated by magnetic separation using a Nd–Fe–B magnet; then, the particles were washed with absolute ethanol and dried in vacuum to constant weight.

Figure 1 shows TEM images and electron diffraction patterns for MNPs **1**, MNPs **2** and material obtained after heating MNPs **1** at 180 °C during 45 h.

According to TEM data, both types of modified nanoparticles have a core–shell structure with an average size of 10 and 13 nm for MNPs **1** and MNPs **2**, correspondingly. The main phase of the cores of the samples is the magnetite, which is confirmed by point and ring reflections in the electron diffraction region [64]. The thickness of the glycerolate shells of MNPs **1** and MNPs **2** is ~2.2 and 2.7 nm, respectively. The size of the initial MNPs was 9–11 nm. Based on the found sizes of MNP **1** and MNP **2** and the thicknesses of their shells, it can be concluded that the MNP **1** cores decrease in size to ~8 nm during modification, while the MNP **2** cores remain practically unchanged. In the first case, the formation of the shell occurs due to the chemical reaction of iron atoms of the core with glycerol molecules. As a result, the core size decreases. A similar process was demonstrated by us in [52]. In the second case, this process is less pronounced, since, in addition to the Fe(III)Glyc shell, a SiGlyc-based shell is formed to a large extent.

Heating MNPs **1** for 45 h in glycerol leads to almost complete conversion to Fe(III)Glyc (Figure 1c). Thus, submicron formations based on Fe(III)Glyc (200–300 nm) with rare inclusions of Fe_3_O_4_ MNPs of various sizes could be observed on TEM images (Figure 1c).

XRD data confirm the presence of the phases of Fe(III)Glyc and magnetite (Fe_3_O_4_) (Figure 2). The diffraction patterns of MNPs **1** and MNPs **2** contain a reflex at 12.7° (2θ), which is the main diffraction band characteristic of Fe(III)Glyc (Powder Diffraction File JCPDSD-ICDD PDF2, for the iron glycerolate phase map [23-1731]). In the XRD pattern of MNPs **2**, the amorphous region in the range of 15.0°–35.0° (2θ) is characteristic of SiGlyc. The diffraction patterns of MNPs **1** and MNPs **2** also show reflections related to the Fe_3_O_4_ phase (Powder Diffraction File JCPDSD-ICDD PDF2, for the magnetite phase map [28-0491]).

In order to characterize the composition of the core and shell of MNPs, we used the Mössbauer spectroscopy method. We used this method to determine the phase composition of the core, as well as to quantify the ratio of Fe in the core and shell. It was found that the phase of the bare MNPs corresponds to the phase of non-stoichiometric magnetite Fe_3_O_4_ (Figure 3a).

The parabolic shape of the background line is associated with the presence of a large fraction of magnetic particles near the superparamagnetic transition (the blocking temperature T_B_) [65]. It is difficult to correctly determine the ratio of iron ions from the spectrum in a zero external field (*H*_ext_) for the powder of the bare nanoparticles (Figure 3a). This is due to the fact that the line intensity is redistributed by the reason of the proximity to the superparamagnetic state. Thus, we observe a large distribution with much smaller *H*_hf_ values instead of hyperfine fields (*H*_hf_) corresponding to the known values for oxides. The application of *H*_ext_ (Figure 3a) transforms the fine fraction into a stable magnetically ordered state, and the corresponding background disappears. The intensity ratio of Fe^2+^/Fe^3+^ in two spectra (see below) indicates the predominance of Fe^3+^ ions in the fine fraction of nanoparticles.

Since iron oxides and the organic compound Fe(III)Glyc have different Debye temperatures, the resonant absorption and the corresponding intensities of the subspectra for the particle core and shell should be different. Therefore, a reference mechanical mixture was prepared containing 30 mg of MNPs Fe_3_O_4_ and 63 mg of Fe(III)Glyc in order to quantify the Fe content in the core and shell.

The intensities of the paramagnetic (doublet) and ferromagnetic (two sextets) contributions from the Fe atoms in the reference mixture of Fe(III)Glyc and Fe_3_O_4_ were related as 70:30, respectively (Figure 3b). The mass ratio of Fe in the same Fe(III)Glyc: Fe_3_O_4_ mixture was 53:47. These data were used to calculate the calibration factor and determine the Fe content in the core (Fe_3_O_4_) and in the glycerolate shell of the modified particles (MNPs **1** and MNPs **2**). Figure 3c,d show the Mössbauer spectra of MNPs **1** and MNPs **2**. The spectra consist of three components. The sextet lines refer to two structural positions of iron atoms in the ferrimagnetic phase of iron oxide Fe_3_O_4_, which belongs to the core of MNPs. The doublet line refers to the paramagnetic phase of the Fe(III)Glyc shell of MNPs according to the values of the isomer shift and quadrupole splitting (Table 1), by analogy with Ref. [51]. It is worth noting that the values of hyperfine fields for both iron positions in Fe_3_O_4_ are lower than in a bulk iron oxide. Likely, this is associated with vacancies in the iron sublattice and, as a consequence, with the non-stoichiometry of Fe^3+^/Fe^2+^ ions as reported in [66,67]. According to [67], non-stoichiometry and vacancies in (Fe^3+^_A-site_[Fe _y_ ^3+ (^Fe _x_ ^2+^Fe _x_ ^3+^) □ _2–2x-y_]_B-site_ O_4_ changes fraction of Fe_2_O_3_ (or Fe^3+^[Fe_5/3_ ^3+^ □_1/3_]O_4_) in the oxide composition (□—vacancies in the Fe sublattice). We have obtained r_AB_ = 1, and Fe^2+^/Fe^3+^ = 0.34, instead of the ratio of the intensities for two crystallographic positions of iron atoms *S*(Fe^2+^)/*S*(Fe^3+^)= r_AB_ = 0.52 for pure stoichiometric magnetite. Based on this, and following [53], we determined the fractions of Fe_3_O_4_ and Fe_2_O_3_ in the core of MNPs **1** nanoparticles as 77 and 23 wt.%, respectively. For the bare MNPs, the intensity ratio of the values was r_AB_ = 1.43; Fe^2+^/Fe^3+^ = 0.27, Fe_3_O_4_ (61 wt.%), Fe_2_O_3_ (39 wt.%). Hyperfine fields, *H*_hf_, for both positions are lower than in bulk by 10 kOe, which is consistent with a larger fraction of Fe^2+^ in the oxide. In MNPs **1** and MNPs **2**, a decrease in the part of Fe^3+^ is observed compared to the bare MNPs. Therefore, Fe^3+^ ions react more actively when obtaining particles with an Fe(III)Glyc shell. This is likely due to the above conclusion that the part of Fe^3+^ prevails in the finer fraction nanoparticles that react more actively due to the larger specific surface area.

Based on the results of processing the spectra and taking into account the calibration by the line intensities, the distribution of Fe in the core and shell of the synthesized materials was calculated, as well as the mass ratio of the shell and core (Table 2).

Using the C,H-elemental analysis data (Table 2), we can calculate the mass fraction of the Fe(III)Glyc shell in MNPs **1** using Equation (1):ω′_Fe(III)Glyc_ = ω_C_ × 100% / ω_C in Fe(III)Glyc_,(1)
where ω′_Fe(III)Glyc_ is the fraction (wt.%) of Fe(III)Glyc shell in MNPs **1**; ω_C_ is C content (wt.%) in MNPs **1** found by the C,H-elemental analysis (Table 2); ω_C in Fe(III)Glyc_ is calculated C content (24.86 wt.%) in Fe(III)Glyc. 

Equations (2) and (3) can be used to find the Fe content in the shell and core of MNPs **1**, respectively (Table 2):ω′_Fe in shell_ = ω′_Fe(III)Glyc_ × ω_Fe in Fe(III)Glyc_/100%(2)
ω′_Fe in core_ = ω_Fe_ − ω_Fe in shell_(3)
where ω^’^_Fe in shell_ is Fe content (wt.%) in Fe(III)Glyc shell of MNPs **1**; ω_Fe in Fe(III)Glyc_ is calculated Fe content (38.54 wt.%) in Fe(III)Glyc; ω_Fe in core_ is Fe content (wt.%) in the core of MNPs **1**; ω_Fe_ is Fe content (wt.%) in MNPs **1** found by the ICP AES (Table 2).

We concluded that the calculation results obtained from the elemental analysis data correlate with the Mössbauer spectroscopy data (Table 2).

It should be noted that, for MNPs **2**, it is impossible to determine the percentage of the Fe(III)/SiGlyc shell in a similar way from elemental analysis data, since the ratio of Fe(III)Glyc and SiGlyc is unknown. The percentage of Fe in the core and mixed shell cannot be calculated from elemental analysis data either. Therefore, the above method of calculating these parameters, based on Mössbauer spectroscopy data, seems to be very valuable for characterizing mixed-shell MNPs.

For the MNPs **2** sample with a mixed shell of iron and silicon glycerolates, according to Mössbauer spectroscopy data, it was determined that the iron atoms in the shell (Fe(III)Glyc) and core (Fe_3_O_4_) related as 32 and 68 at.%. Taking into account the Fe content determined for the same sample by the ICP AES method, it is possible to estimate the fraction of Fe(III)Glyc (wt.%) in the composition of the mixed shell according to Equation (4), as well as the fraction of the core (wt.%) according to Equation (5) (Table 2).
ω″_Fe(III)Glyc_ = ω″_Fe in shell_ × ω_Fe_/ω_Fe in Fe(III)Glyc_(4)
where ω″_Fe(III)Glyc_ is Fe(III)Glyc content (wt.%) in the shell of MNPs **2**; ω″_Fe in shell_ is Fe content (wt.%) in Fe(III)/SiGlyc shell of MNPs **2** found by the Mössbauer spectroscopy; ω_Fe_ is Fe content (wt.%) in MNPs **2** found by the ICP AES (Table 2); ω_Fe in Fe(III)Glyc_ is calculated Fe content (38.54 wt.%) in Fe(III)Glyc.
ω″_Fe3O4_ = ω″_Fe in core_ × ω_Fe_/ω_Fe in Fe3O4_(5)
where ω″_Fe3O4_ is Fe_3_O_4_ core fraction (wt.%) in MNPs **2**; ω″_Fe in core_ is Fe content (wt.%) in core of MNPs **2** found by the Mössbauer spectroscopy; ω_Fe in Fe3O4_ is calculated Fe content (72.36 wt.%) in Fe_3_O_4_.

The fraction of SiGlyc (ω″_SiGlyc_) in the composition of the mixed shell of MNPs **2** can be found by Equation (6) (Table 2): ω″_SiGlyc_ = 100% − ω″_Fe(III)Glyc_ − ω″_Fe3O4_(6)

The ratio of Fe and Si atoms in the Fe(III)/SiGlyc mixed shell, based on Mössbauer spectroscopy and ICP AES data, corresponds to ~1:1. Taking into account the elemental analysis data, we determined the ratio of Si: Glyc groups as 1:2, which indicates the formation of a SiGlyc shell of the composition indicated in Figure 1. Thus, the mixed shell of MNPs **2** consists of two components: Fe(III)Glyc and SiGlyc. Fe(III)Glyc is formed as a result of the interaction of Fe_3_O_4_ with glycerol; in this case, the process is accompanied by the oxidation of Fe(II) to Fe(III) by air oxygen and the release of H_2_O during the condensation of iron oxides with glycerol. As noted above, Fe(III) ions more actively react with glycerol when obtaining particles with a Fe(III)Glyc shell. SiGlyc are formed as a result of the partial hydrolysis of Si(C_3_H_7_O_3_)_4_ added to the reaction followed by the condensation of silanol groups to form Si–O–Si groups containing residual glyceroxy groups at the Si atom in the 3D polymer network. Based on the elemental analysis data using Equation (7), it was calculated that MNPs **1** and MNPs **2** contain 2.56 mmol and 3.75 mmol of glycerol residues (C_3_H_5_O_3_) per 1 g of MNPs, respectively (by analogy with [68]) (Table 2).c_Glyc_ = ω_C_ / (ω_C in Glyc_ × *M*)(7)
where ω_C_ is C content (wt.%) in MNPs found by the C,H-elemental analysis; ω_C in Glyc_ is C content (wt.%) in glycerolate residues (C_3_H_5_O_3_), 45.41%; *M* is the molar mass of the glycerolate residues (C_3_H_5_O_3_), 89.07 g/mol.

The fraction of the organic shell was also estimated using thermogravimetry analysis (TGA) (Figure 4a–c).

The mass loss of MNP samples at temperatures up to 100 °C can be associated with the removal of physically adsorbed water from their surface, which is confirmed by the data of the TG-IR analysis (FT-IR analysis showed the presence of only H_2_O in the evolved gases) flow (Figure 4b,c) by analogy with [36,69]). MNPs **1** and MNPs **2** contained <0.5% (as in [38]) and 2.6% of physically adsorbed H_2_O, respectively (Figure 4a). Based on the presence of maxima on the DTG curve, we concluded that the decomposition of the organic coating of MNPs **1** and MNPs **2** samples occurs in three main stages (1—up to 240 °C, 2—240–500 °C (for MNPs **1**) or 240–560 °C (for MNPs **2**), and 3—up to 900°C). Using the example of a TG-IR analysis of the composition of evolved gases after heat treatment of MNPs **1**, we have shown that, at the first stage (up to 240 °C (~21 min)), an active evolution of H_2_O (maximum in the H_2_O evolution profile), CO_2_, as well as a small amount of CO, takes place (Figure 4b,c).

This may be due to the removal of hydroxyl groups from the surface of MNPs [36] and, probably, to the decomposition of CH–OH or CH_2_–OH fragments of glycerolates. At the next stage, the thermal destruction of the carbon skeleton of glycerolates occurs. Thus, at 26 min (~300 °C), the main maximum in CO_2_ emission profile, as well as small amounts of CO and H_2_O, are observed (Figure 4b,c). This reaction was accompanied by a pronounced exothermic effect (with a maximum at ~320 °C (Figure 4a)), which is characteristic of both types of materials. At the third stage, the carbonization of residues of organic molecules on the MNP surface occurred, which was accompanied by the release of CO_2_ (CO was observed in trace amounts, traces of H_2_O were absent) (Figure 4b,c). The total mass loss of samples MNPs **1** and MNPs **2** due to the decomposition of glycerol residues was 31.3 and 39.0%, respectively.

The presence of a glycerolate shell in modified MNPs is also confirmed by IR spectroscopy data. Figure 5 shows the IR spectra of MNPs **1** and MNPs **2**, as well as Fe(III)Glyc and silicon glycerolates.

Intense bands at 2853–2923 cm^–1^ in the spectra of MNPs **1** and MNPs **2** correspond to the stretching vibrations of C–H bonds, and bands with maxima in the range of 1221–1451 cm^–1^ correspond to the bending vibrations of C–H bonds (wagging, twisting and scissoring vibrations) in the CH and CH_2_ of glycerolates. The absorption bands at 822–1119 and 505–713 cm^–1^ correspond to the C–O stretching and bending vibrations in the C–O–Fe groups of the glycerolates. The broadened bands at 3323–3335 and 1593–1601 cm^–1^ indicate the presence of physically adsorbed water molecules in MNPs **1** and MNPs 2 (the intensity of these bands in the spectrum of MNPs **2** is noticeably higher than for MNPs 1). The band at 537 cm^–1^ is a characteristic band for the Fe–O vibrations of the initial MNPs [37]. In the modified products, it shifts to the region of 577 and 580 cm^–1^ (for the spectra of MNPs **1** and MNPs **2**, respectively) and this band is superimposed on the bands in the region of 502–635 cm^–1^, corresponding to the C–O vibrations of Fe(III)Glyc. The IR spectrum of MNPs 2, in addition to those described above, also contains broadened absorption bands in the region of 960–1120 cm^–1^, which are characteristic of the initial silicon glycerolates (the bands at 998, 1031 and 1107 cm^–1^, corresponding to both C–O stretching vibrations in C–O–H of the glycerol residue, as well as Si–O and Si–O–Si).

### 2.2. Evaluation of Magnetic Properties of MNPs **1** and MNPs **2**

The synthesized materials had high values of saturation magnetization (*M*_S_), which is due to a small proportion of their glycerolate shell, as well as low coercive force (*H*_C_) (up to 5 Oe) and low remanence magnetization (*M*_R_) (up to 0.5 emu/g) (Figure 6a). 

It has been demonstrated that Fe(III)Glyc exhibits paramagnetic properties (Figure 6a). Taking into account the *M*_S_ of the initial MNPs (70 emu/g) and the mass fractions of the shells (Table 2), the calculated *M*_S_ for MNPs **1** and MNPs **2** were 41 and 24 emu/g, respectively, and were close to the values determined by the VMS method (Figure 6a). The decrease in *M*_S_ of the synthesized MNPs only by a value proportional to the paramagnetic shell indicates that the shell does not affect the spin state of surface iron atoms, which could cause a surface spin canting [70,71] and, accordingly, an additional decrease in the *M*_S_ of the material.

Due to the ability to heat up to temperatures above 42 °C in an alternating magnetic field (AMF), magnetic nanoparticles are currently often used as therapeutic agents or delivery vehicles for anticancer drugs with magnetically mediated release. MNPs **1** and MNPs **2** synthesized in this work were shown to be able to rapidly heat up to temperatures of 42 °C (and higher) in 85 and 110 s, correspondingly, under AMF application at the field parameter *H* × *f* (1.8 × 10^7^ Oe Hz; a magnetic field *H* = 192 Oe, frequency *f* = 93.5 kHz), which is less than the safety limit 6.25 × 10^7^ Oe Hz [28,72] (Figure 6b). The values of specific absorption rate (SAR) and intrinsic loss power (ILP) of MNPs **1** and MNPs **2** were calculated for their suspensions at a concentration of 80 mg/mL by analogy with [31] (Figure 6b).

### 2.3. Evaluation of the Hydrolysis of MNPs **1** and MNPs **2** Shells in Aqueous Media

It is known that water-soluble silicon glycerolates (silicon tetraglycerolate) easily undergo hydrolysis with the formation of silanol groups and their subsequent condensation with the formation of Si–O–Si groups in the spatial network of the polymer phase (sol–gel process) [73]. Water-insoluble Fe(III)Glyc is more resistant to hydrolysis [51].

In this work, we studied the possibility of hydrolysis of Fe(III)Glyc, as well as shells of MNPs **1** and MNPs **2** in water and in a 72:28 H_2_O: glycerol mixture. For this, the data of IR spectroscopy and C,H-elemental analysis were used. It was shown that Fe(III)Glyc and the MNPs **1** shell in aqueous media do not undergo hydrolysis after being suspended in water (1.0 mg/mL) for 24 h at 25 °C (no changes were observed in the IR spectra (Figure 7a), and the wt.%C in the samples did not change (see Section 3.7). 

In the IR spectra of MNPs **2** (1.0 mg/mL), one can notice a significant decrease in the intensity of absorption bands related to stretching (2858 cm^–1^) and bending (1320–1450^–1^) vibration C–H bonds of glycerolates. However, the spectra of MNPs **2** retain bands characteristic of Fe(III)Glyc (824, 713 cm^–1^) and Fe_3_O_4_ (589 cm^–1^). Si–O–Si (1057, 1006, 959 cm^–1^), on the contrary, became more intense, probably due to an increase in the content of this type of bonds on the surface of nanoparticles due to the hydrolysis of silicon glycerolates in the composition of the MNPs **2** shell. For MNPs **2,** the hydrolysis kinetics of glycerolate shell under the conditions described above was additionally studied. For samples taken from their aqueous suspension after 1, 3, 6, 10 and 24 h and washed with water, a decrease in wt. % C in the samples was observed (Figure 7b). This indicates the occurrence of the hydrolysis of the glycerolate shell. Thus, the hydrolysis of the MNPs **2** shell for 24 h at 25 °C was 27.5%, which may be due to the hydrolysis of the SiGlyc component of the shell forming 36% of the total mass of MNPs **2** (see Table 2). It is known that an excess of glycerol in the system prevents this process [74]. However, as we have shown, the use of a mixture of H_2_O: glycerol (72:28) or an increase in the concentration of the MNP solution from 1.0 to 10 mg/mL did not affect the degree of hydrolysis (see Section 3.7, Table 3).

Thus, we have shown that if Fe(III)Glyc is resistant to hydrolysis, then SiGlyc in the MNPs **2** shell is rather labile and hydrolyzes by 76.4% of the initial SiGlyc content in the shell for 24 h at 25 °C.

### 2.4. MTT Cytotoxicity Assay

The studied MNPs **1** and MNPs **2** did not show a statistically significant effect on Vero cell culture for 48 h in the MTT assay at any of the studied concentrations (0.01–1.0 mg/mL) (Figure 8) and can therefore be considered as non-toxic.

## 3. Materials and Methods

### 3.1. Materials

We used FeCl_3_ × 6H_2_O and FeSO_4_ × 7H_2_O (Vekton, St. Petersburg, Russia), tetraethoxysilane (TEOS, Vekton, St. Petersburg, Russia) and glycerol (Vekton, St. Petersburg, Russia).

### 3.2. Synthesis of MNPs 

A saturated solution of NH_4_OH (5 mL) was added to 45 mL of an aqueous solution of FeCl_3_ × 6H_2_O (1.051 g, 3.89 mmol) and FeSO4 × 7H_2_O (0.540 g, 1.94 mmol) under sonication on US-bath at 40 °C (by analogy with Refs. [36,75]). After 10 min, nanoparticles were precipitated with a Nd-Fe-B magnet, washed with H_2_O to a neutral pH and then with EtOH 5 × 20 mL. The obtained MNPs were dried under reduced pressure at 25 °C.

### 3.3. Synthesis of MNPs with Iron III Glycerolate Shell (MNPs **1**)

Suspension of MNPs (0.86 g, 3.7 mmol) in dry glycerol (27.80 g, 0.302 mol) was stirred at 180 °C for 18 h. The particles were precipitated with a Nd-Fe-B magnet, washed with EtOH (abs) 5 × 20 mL and dried under reduced pressure at 25 °C to yield 0.99 g of MNPs **1**. Elemental analysis, Found: C, 10.36; H, 1.60. Fe, 57.56. IR, ν/cm^–1^: 3335 (ν(O−H), H_2_O); 2922, 2853 (ν(C−H), CH_2_, CH); 1593 (δ (O−H), H_2_O); 1447, 1378, 1322,1303, 1251, 1221 (δ(C−H), CH_2_, CH); 1119, 1089, 1055, 1002, 959, 914, 858, 823, 713, 577, 505 (ν(C−O), δ(C−O−Fe); γ(C−C); ν(Fe−O), Fe_3_O_4_ core). Mössbauer: Fe^3+^ doublet, δ_iso_ = 0.66 mm/s, *Q*_S_ = 0.51 mm/s.

### 3.4. Synthesis of Silicon Tetraglycerolate Si(C_3_H_7_O_3_)_4_


Silicon glycerolates in glycerol (6 molar excess) were obtained according to a previously researched procedure [76] by transesterification of Si(OC_2_H_5_)_4_ with glycerol in a molar ratio of Si(OC_2_H_5_)_4_: C_3_H_8_O_3_ = 1:10 at 130 °C for 3 h (before applying Si(OC_2_H_5_)_4_ and were distilled at atmospheric pressure; glycerol was distilled under reduced pressure). The resulting EtOH was removed first at atmospheric pressure, then under reduced pressure on a rotary evaporator to constant weight. The synthesized product was a colorless transparent viscous liquid, easily soluble in water, *n*_D_^20^ 1.4815. The composition of the product corresponded to the molar ratio Si(C_3_H_7_O_3_)_4_: C_3_H_8_O_3_ = 1:6. The results of elemental analysis and IR spectroscopy are consistent with the data of in ref. [76].

### 3.5. Synthesis of MNPs with Iron and Silica Glycerolates Shell (MNPs **2**)

MNPs (0.86 g, 3.7 mmol) were dispersed in silicon tetraglycerolate Si(C_3_H_7_O_3_)_4_ in 6-mol excess of glycerol (27.80 g, 0.029 mol). The reaction mixture was stirred at 180 °C for 18 h. The particles were separated by a Nd-Fe-B magnet, washed with EtOH (abs) 5 × 20 mL and dried under reduced pressure at 25 °C to yield 1.50 g of MNPs **2**. Elemental analysis, Found: C, 15.16; H, 2.80; Fe, 35.88; Si, 5.33. IR, ν/cm^–1^: 3323 (ν(O−H), H_2_O, C−O−H), 2923, 2855 (ν(C-H), CH_2_, CH); 1601 (δ (O−H), H_2_O); 1451, 1379, 1323, 1304, (δ(C-H), CH_2_ and CH); 1117, 1085, 1051, 1002, 959, 914, 859, 822, 713, 580, 505 (ν(C−O), δ(C-O-Fe), γ(C-C), ν(Fe-O)). Mössbauer: Fe^3+^ doublet, δ_iso_ = 0.39 mm/s, *Q*_S_ = 0.51 mm/s; Fe^3+^ sextet, δ_iso_ = 0.37 mm/s, *H*_hf_ = 462 kOe; Fe^2+^ sextet, δ_iso_ = 0.44 mm/s, *H*_hf_ = 420 kOe.

### 3.6. Synthesis of Fe(III)Glyc

FeCl_3_·6H_2_O (4.00 g, 14.798 mmol) and NaOH (1.78 g, 44.4 mmol) were added to anhydrous glycerol (50 mL) by analogy with [51]. The reaction mixture was heated under magnetic stirring at 180 °C for 18 h, then poured into distilled H_2_O (100 mL) and stirred on a magnetic stirrer for 15 min. The precipitate was filtered off, washed with distilled H_2_O (100 mL) and EtOH (25 mL) and oven-dried at 100 °C for 6 h to yield 1.95 g (91%) of Fe(III)Glyc. Elemental analysis, Calculated for FeC_3_H_5_O_3_: C, 24.86; H, 3.48; Fe, 38.54; Found: C, 24.70; H, 3.43; Fe, 38.40. IR, ν/cm^–1^: 2925, 2859 (ν(C−H), CH_2_, CH); 1460, 1445, 1324, 1248 (δ(C−H), CH_2_ and CH); 1119, 1091, 1057, 1004, 971, 955, 914, 857, 820, 707, 604, 502 (ν(C−O), δ(C−O−Fe), γ(C−C)). Mössbauer: Fe^3+^ doublet, δiso = 0.66 mm/s, QS = 0.48 mm/s [51].

### 3.7. Hydrolysis of Fe(III)Glyc, MNPs **1** and MNPs **2** Shell

Hydrolysis of Fe(III)Glyc and modified MNPs was carried out at a concentration of 1.0 or 10 mg/mL in aqueous or aqueous glycerol (28% glycerol) media (Table 3). The dispersion was stirred for 24 h at 25 °C; the material was precipitated with a Nd–Fe–B magnet (in the case of MNPs) or by centrifugation (in the case of Fe(III)Glyc), washed with EtOH (abs.) 5 × 20 mL and dried under reduced pressure. The materials were characterized by the data of IR spectroscopy and CH-elemental analysis (Table 3). 

The hydrolysis kinetics of the MNP **2** shell was carried out using an aqueous suspension of nanoparticles at a concentration of 1.0 mg/mL at 25 °C. Aliquots were taken at 1, 3, 6, 10, 24 h and processed as described above.

### 3.8. Characterization of Nanoparticles

Transmission electron microscopy (TEM) images were obtained on a Jeol Jem 2100 (Jeol, Tokyo, Japan) equipped with an Olympus Cantaga G2 digital camera and an Oxford Inca Energy TEM 250 microanalysis system, at 200 kV and 105 mA. XRD was performed on a Shimadzu XRD 700 diffractometer (Shimadzu, Tokyo, Japan) with Cu-Kα radiation. Mössbauer spectra were recorded using an improved MS-2201 spectrometer [77] with a ^57^Fe(Cr) resonant detector in transmission geometry at a temperature of 295 K. The source of γ-radiation was the ^57^Co(Cr) isotope with an activity of 30 mCi. Experimental spectra were processed using Unifem MS software. Pure iron was used as the standard for calibration. Samples were prepared by deposition from a solution of ethanol and polyvinyl butyral glue onto aluminum foil. The content of Si and Fe (wt.%) was determined by the ICP AES on an iCAP 6300 Duo optical emission spectrometer (Thermo Scientific, Waltham, MA, USA). C,H-elemental analysis was carried out using a EuroEA 3000 automatic analyzer (EuroVector, Instruments & Software, Milan, Italy). The IR spectra were recorded on a Perkin Elmer Spectrum Two FT-IR spectrometer (Perkin Elmer, Waltham, MA, USA) equipped with the ultra-attenuated total reflection (UATR) accessory on the diamond crystal. The magnetic properties were studied on a vibrating-sample magnetometer (H up to 25 kOe at 25 °C). The heat release of suspensions of the obtained materials was measured in a solenoid with a field of 192 Oe at a frequency of 93.5 kHz. SAR and ILP were calculated using Formulas (8) and (9), respectively.
SAR = d*T*/d*t*·*m*/*m*_Fe_·*C*(8)
ILP = SAR/(*H*^2^ *f*)(9)
where d*T*/d*t* is the sample heating rate for the first 60 s, which was determined by the slope of the initial section of the suspension heating curve after the magnetic field was switched on, K/s; *m* is the suspension weight, g; *m*_Fe_ is the mass of nanoparticles in suspension, g; *C* is the specific heat capacity of the suspension, J/g·K; *H* and *f* are AMF field amplitude (Oe) and frequency (Hz), correspondingly.

### 3.9. Assessment of Cytotoxicity of MNPs **1** and MNPs **2**

#### 3.9.1. Cell Cultures

We used Vero cell cultures (green monkey kidney epithelium) obtained from the Russian collection of cell cultures of the Institute of Cytology RAS. 

The cells were cultured in T-25 ventilated culture bottles (JetBiofil, Guangzhou, China) in DMEM nutrient medium (HiMedia, Mumbai, India) with addition of 3% fetal calf serum (Biolot, Moscow, Russia) and gentamicin–streptomycin solution (Biolot, Moscow, Russia). The cells were maintained in an incubator with a humidified atmosphere with 5% CO_2_. 

Cells were passaged every three days (or when 90% confluence was reached) using Trypsin-Versen solution (ServiceBio, Wuhan, China).

#### 3.9.2. Preparation of Samples of MNPs **1** and MNPs **2**

Suspensions of two types of nanoparticles placed in 1.5 mL plastic centrifuge microtubes were used for the study. The required volumes were taken using a pipette and mixed with complete DMEM nutrient medium to obtain concentrations of 1.0, 0.10 and 0.01 mg/mL.

All manipulations with the substances were performed as rapidly as possible and under aseptic conditions and the solutions of the substances were added to the cells immediately after preparation.

#### 3.9.3. MTT Assay 

The MTT test is an available test for screening the cytotoxicity of various substances on cell cultures [78]. This method is based on the study of mitochondrial activity associated with cell viability. Under normal conditions, mitochondrial cell enzymes are capable of reducing the yellow tetrazolium dye 3-(4,5-dimethylthiazol-2-yl)-2,5-diphenyl-tetrazolium bromide into insoluble formazan, which has purple staining.

For the study, cells were plated into a 96-well plate (JetBiofil, Guangzhou, China) and grown to 70% monolayer, after which the medium was taken with a multichannel pipette and replaced with prepared medium with the addition of the test substance. 

Incubation with the test substance was carried out for 24 h, after which the medium was removed and replaced with complete nutrient medium with MTT added (at a concentration of 1 mg/mL), followed by incubation for 2 h. Then, the medium was removed and 100 µL of DMSO was poured into the wells of the plate. After complete dissolution of the dye, the staining intensity was assessed using a plate photometer at a wavelength of 570 nm. Since the nanoparticles themselves have a fairly appreciable optical density at this wavelength, this value was subtracted from the optical density of the dye after the procedure.

#### 3.9.4. Statistical Processing

The results were analyzed with a python script, available in the repository at: https://github.com/arteys/MTT_assay_multi (accessed on 30 June 2023). Raw data are also given in this repository. Statistical processing was performed using Statannotations library [https://github.com/trevismd/statannotations] (accessed on 30 June 2023), using Kruskal–Wallis one-way analysis of variance test, with Bonferroni correction for multiple comparisons [79].

## 4. Conclusions

Novel magnetic nanocomposite materials based on Fe_3_O_4_ nanoparticles coated with Fe(III)Glyc or Fe(III)/SiGlyc were obtained. The synthesized nanocomposites were characterized using TEM, XRD, TGA, VMS, Mössbauer and IR spectroscopy. Both types of modified nanoparticles have a core–shell structure with an average core size of 10 and 13 nm and shell size of ~2.2 and 2.7 nm for MNPs coated with Fe(III)Glyc or Fe(III)/SiGlyc, respectively. The amounts of iron and silica glycerolates in the nanocomposites were calculated with Mössbauer spectroscopy, ICP AES and C,H-elemental analysis. As a result, shell: core weight ratios were calculated to be 41:59 and 66:34 for MNPs coated with Fe(III)Glyc and Fe(III)/SiGlyc mixed, respectively. The synthesized nanocomposites had high specific magnetization values and a high magnetic response to the alternating magnetic field. It was shown that, while the Fe(III)Glyc is resistant to hydrolysis, the SiGlyc in the composition of the Fe(III)/SiGlyc mixed shell is rather labile and hydrolyzes by 76.4% of the initial content of SiGlyc in the shell for 24 h at 25 °C. The hydrolysis of glycerolate shells in aqueous solutions over time can contribute to the slow desorption of drugs, providing their prolonged release. The synthesized materials have shown no cytotoxicity in in vitro experiments (MTT-assay). Thus, we believe that the data obtained can be used in the design of materials for the delivery of drugs with controlled release.

## Data Availability

Not applicable.

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
