# Peer review of "Magnetic Nanocomposite Materials Based on Fe3O4 Nanoparticles with Iron and Silica Glycerolates Shell: Synthesis and Characterization"

_ijms, 2023, doi:10.3390/ijms241512178_

Round 1

Reviewer 1 Report

An article describing a method for the synthesis of new magnetic nanocomposites based on Fe3O4 nanoparticles is submitted for review. The paper presents a large set of modern research methods, including TEM, XRD, TGA, VMS, Mössbauer and IR spectroscopy, Bauer spectroscopy, ICP AES and C,H-element analysis. The ratio of the core to the shell is established, and the resistance of the system to hydrolysis is shown. It has been established that the particles do not exhibit cytotoxicity and can be recommended for controlled release drug delivery.

The article has all the signs of novelty and is recommended for publication.

Questions:

1. It is necessary to indicate the size of the initial MNPs used in the synthesis of nanocomposites. Based on the given sizes of MNPs 1 and MNPs 2 and the thicknesses of their shells, it turns out that the MNPs 1 cores have an average diameter of 7.8 nm, and the MNPs 2 cores have an average diameter of 10.3 nm. How can this be explained?

2. What does “□” stand for in the Mössbauer spectroscopy data description section?

3. It is not clear from the text of the manuscript how glycerolate residues (С3Н5О3) per 1 g of MNPs were calculated. Please provide an equation for the calculation.

Author Response

Questions:

  1. It is necessary to indicate the size of the initial MNPs used in the synthesis of nanocomposites. Based on the given sizes of MNPs 1 and MNPs 2 and the thicknesses of their shells, it turns out that the MNPs 1 cores have an average diameter of 7.8 nm, and the MNPs 2 cores have an average diameter of 10.3 nm. How can this be explained?

Reply: Thank you very much for your appreciation of our work. The size of the initial MNPs was 9–11 nm. Based on the found sizes of MNP 1 and MNP 2 and the thicknesses of their shells, it can be concluded that the MNP 1 cores decrease in size to ~8 nm during modification, while the MNP 2 cores remain practically unchanged. In the first case, the formation of the shell occurs due to the chemical reaction of iron atoms of the core with glycerol molecules. As a result, the core size decreases. A similar process was demonstrated by us in [Demin, A.M.; Khonina, T.G.; Shadrina, E.V.; Bogdanova, E.A.; Kuznetsov, D.K.; Shur, V.Ya.; Krasnov, V.P. Synthesis of nanocomposite with a core-shell structure based on Fe3O4 magnetic nanoparticles and iron glycerolate. Russ. Chem. Bull. (Int. Ed.) 2019, 6, 1178–1182. doi.org/10.1007/s11172-019-2536-x]. In the second case, this process is less pronounced, since, in addition to the Fe(III)Glyc shell, a SiGlyc-based shell is formed to a large extent.

  1. What does “□” stand for in the Mössbauer spectroscopy data description section?

Reply: ‘□ – vacancies in the Fe sublattice’. Information added to the text of the manuscript.

  1. It is not clear from the text of the manuscript how glycerolate residues (С3Н5О3) per 1 g of MNPs were calculated. Please provide an equation for the calculation.

Reply: Information added to the text of the manuscript.

Reviewer 2 Report

This is a rather good and important research article, and it can be recommended for publication after clarification / improvement of some ambiguities.

1.     In the Introduction, it is required to clarify what role the size of nanoparticles plays.

2.     To attract a wider readership and visibility of this work, more simple and understandable information about the oxide would be helpful. Speaking of Fe3O4 oxide nanoparticles, the role and effects of their sizes should also be mentioned. See, for example, few recent MDPI papers and references therein:

Serga, V et al. Impact of Gadolinium on the Structure and Magnetic Properties of Nanocrystalline Powders of Iron Oxides Produced by the Extraction-Pyrolytic Method. Materials 2020, 13, 4147. https://doi.org/10.3390/ma13184147

Wang, D.; et al. Ni-Pd-Incorporated Fe3O4 Yolk-Shelled Nanospheres as Efficient Magnetically Recyclable Catalysts for Reduction of N-Containing Unsaturated Compounds. Catalysts 202313, 190. https://doi.org/10.3390/catal13010190

Liu, M., Ye, Y., Ye, J., Gao, T., Wang, D., Chen, G., & Song, Z. (2023). Recent Advances of Magnetite (Fe3O4)-Based Magnetic Materials in Catalytic Applications. Magnetochemistry, 9(4), 110.

3.     In this form, the introduction looks very special and abstruse and informative only for a narrow number of specialists. So, an additional paragraph on the recent advances for obtaining/applications of  Fe3O4 would be extremely useful.

4.     Was there any modification of the nanoparticles/structures during TEM/electron diffraction measurements? All silica-based materials are not radiation resistant and uncontrolled surface modification can be expected.

5.     How does Fe–O vibrations frequency depend on the size of nanoparticles?

6.     The conclusions must be modified. Speaking of nanoparticles and keeping silent about their size is not entirely correct. Be more clear about what fundamentally new data you have obtained

Author Response

  1. In the Introduction, it is required to clarify what role the size of nanoparticles plays.

Reply: Thank you very much for your appreciation of our work. Information added to the text of the manuscript (Introduction).

  1. To attract a wider readership and visibility of this work, more simple and understandable information about the oxide would be helpful. Speaking of Fe3O4oxide nanoparticles, the role and effects of their sizes should also be mentioned. See, for example, few recent MDPI papers and references therein:

Serga, V et al. Impact of Gadolinium on the Structure and Magnetic Properties of Nanocrystalline Powders of Iron Oxides Produced by the Extraction-Pyrolytic Method. Materials 2020, 13, 4147. https://doi.org/10.3390/ma13184147

Wang, D.; et al. Ni-Pd-Incorporated Fe3O4 Yolk-Shelled Nanospheres as Efficient Magnetically Recyclable Catalysts for Reduction of N-Containing Unsaturated Compounds. Catalysts 202313, 190. https://doi.org/10.3390/catal13010190

Liu, M., Ye, Y., Ye, J., Gao, T., Wang, D., Chen, G., & Song, Z. (2023). Recent Advances of Magnetite (Fe3O4)-Based Magnetic Materials in Catalytic Applications. Magnetochemistry, 9(4), 110.

Reply: Information added to the text of the manuscript (Introduction).

  1. In this form, the introduction looks very special and abstruse and informative only for a narrow number of specialists. So, an additional paragraph on the recent advances for obtaining/applications of  Fe3O4would be extremely useful.

Reply: Information added to the text of the manuscript (Introduction).

  1. Was there any modification of the nanoparticles/structures during TEM/electron diffraction measurements? All silica-based materials are not radiation resistant and uncontrolled surface modification can be expected.

Reply: We did not observe any visible changes during the analysis of MNPs by the TEM method for both the core and the shell.

  1. How does Fe–O vibrations frequency depend on the size of nanoparticles?

Reply: The band in the range of 530-580 cm–1 is a characteristic band for the Fe–O vibrations (stretching mode of the tetrahedral sites) of magnetite nanoparticles. Based on our experimental data, as well as on the basis of the analysis of literature data related to the synthesis of MNPs Fe3O4 by various methods [J. Phys. Chem. Solid., 2022, 169, 110855; Scientific Reports, 2021, 118643; Ind. Eng. Chem. Res. 2020, 59, 38, 16669–16683; Journal of Spectroscopy, 2018, 2018, 1412563; Langmuir, 2018, 34, 4640–4650], we can conclude that the position of this band is not significantly determined by the size of nanoparticles, but it largely depends on the nature of their coating and the degree of clustering.

As we noted, for example, in the article [Demin A.M., Valova M.S., Pershina A.G., Krasnov V.P. Immobilization of fluorescent protein TagGFP2 on Fe3O4-based magnetic nanoparticles // Russ. Chem. Bull., 2019, 5, 1088–1095 (https://doi.org/10.1007/s11172-019-2524-1)], the band at 532 cm-1, which is characteristic of Fe–O vibrations in the initial MNPs, shifted to the region of 550 cm-1 after protein modification (the core size of MNPs did not change). In our other work [A.M. Demin, A.V. Mekhaev, O.F. Kandarakov, V.I. Popenko, O.G. Leonova, A.M. Murzakaev, D.K. Kuznetsov, M.A. Uimin, A.S. Minin, V. Ya. Shur, A.V. Belyavsky, V.P. Krasnov L-Lysine-modified Fe3O4 nanoparticles for magnetic cell labeling // Colloids Surf., B 2020, 190, 110879. https://doi.org/10.1016/j.colsurfb.2020.110879], modification of MNPs obtained by various methods and having various sizes was carried out; the IR spectra of initial particles with sizes of ~30 and 10 nm contained similar bands (at 540 and 534 cm-1, respectively); upon L-Lys modification, they shifted to the region of 540-550 cm-1. Accurate conclusions can be drawn from experiments with a number of nanoparticles of the same shape, but with different sizes.

  1. The conclusions must be modified. Speaking of nanoparticles and keeping silent about their size is not entirely correct. Be more clear about what fundamentally new data you have obtained.

Reply: Conclusions corrected.

On behalf of all coauthors,

Sincerely yours,

Dr. Alexander M. Demin

Reviewer 3 Report

This study synthesized a novel magnetic nanocomposite based on Fe3O4 nanoparticles (MNP) 465 iron(III) and glycerol coated with silicaglycerol. The physical properties were analyzed by Mössbauer spectroscopy, XRD, FT-IR, Magnetic property, ICP, component analysis, TGA, etc. This demonstrates that the hypothetical compound is validated and not toxic. I have confirmed that there is no problem in most of them, but I think that □ in line 152 is missing some molecular symbol.

Author Response

Reply: Thank you very much for your appreciation of our work.

‘□ – vacancies in the Fe sublattice’. Information added to the text of the manuscript.

On behalf of all coauthors,

Sincerely yours,

Dr. Alexander M. Demin

Round 2

Reviewer 2 Report

The authors have successfully improved the original version of their manuscript, responding constructively to all the comments/recommendations of the reviewer.  Therefore, the article can be recommended for publication.